# Morphine Analgesia, Cannabinoid Receptor 2, and Opioid Growth Factor Receptor Cancer Tissue Expression Improve Survival after Pancreatic Cancer Surgery

**DOI:** 10.3390/cancers15164038

**Published:** 2023-08-09

**Authors:** Lubomir Vecera, Petr Prasil, Josef Srovnal, Emil Berta, Monika Vidlarova, Tomas Gabrhelik, Pavla Kourilova, Martin Lovecek, Pavel Skalicky, Jozef Skarda, Zdenek Kala, Pavel Michalek, Marian Hajduch

**Affiliations:** 1Department of Emergency Medicine, The Tomas Bata Regional Hospital in Zlin, 762 75 Zlin, Czech Republic; lubomir.vecera@bnzlin.cz; 2Department of Paediatric Anaesthesiology and Intensive Care Medicine, University Hospital Brno, Medical Faculty of Masaryk University, 625 00 Brno, Czech Republic; 3Department of Anesthesiology and Intensive Medicine, Landesklinikum Amstetten, 3300 Amstetten, Austria; prasil@seznam.cz; 4Institute of Molecular and Translational Medicine, Faculty of Medicine and Dentistry, Palacky University Olomouc, 779 00 Olomouc, Czech Republic; monika.vidlarova@upol.cz (M.V.); pavla.kourilova@upol.cz (P.K.); marian.hajduch@upol.cz (M.H.); 5Laboratory of Experimental Medicine, Olomouc University Hospital, 779 00 Olomouc, Czech Republic; 6Department of Anaesthesia and Intensive Care, Ringerike Hospital, 3511 Hønefoss, Norway; 7Department of Anaesthesiology, Resuscitation and Intensive Care, The Tomas Bata Regional Hospital in Zlin, 762 75 Zlin, Czech Republic; tomas.gabrhelik@bnzlin.cz; 8Department of Surgery I, University Hospital Olomouc, Faculty of Medicine and Dentistry, Palacky University, 779 00 Olomouc, Czech Republic; martin.lovecek@fnol.cz (M.L.); pavel.skalicky@fnol.cz (P.S.); 9Institute of Molecular and Clinical Pathology and Medical Genetics, Faculty of Medicine, University Hospital Ostrava, University of Ostrava, 703 00 Ostrava, Czech Republic; jozef.skarda@fno.cz; 10Department of Surgery, Faculty of Medicine, University Hospital Brno, Masaryk University, 625 00 Brno, Czech Republic; kala.zdenek@fnbrno.cz; 11Department of Anesthesiology and Intensive Medicine, General University Hospital, First Medical Faculty of the Charles University, 128 00 Prague, Czech Republic; pavel.michalek@vfn.cz; 12Cancer Research Czech Republic Foundation, 779 00 Olomouc, Czech Republic

**Keywords:** pancreatic cancer, pancreatic surgery, morphine, piritramide, postoperative analgesia, cancer recurrence, opioid receptors, cannabinoid receptors, patient survival

## Abstract

**Simple Summary:**

Pancreatic carcinomas are among the most aggressive cancers and have a poor prognosis. The influence of postoperative analgesia on the prognosis of these patients has recently drawn considerable attention. We therefore conducted a retrospective study to evaluate the effect of postoperative analgesia on cancer-specific survival among patients after radical surgery for pancreatic cancer. We also investigated the effect of opioid and cannabinoid receptor gene expressions on overall survival. Our results showed that cancer-specific survival was increased by postoperative analgesia with morphine, cannabinoid receptor 2, and opioid growth factor receptor cancer tissue gene expressions but was reduced by delta opioid receptor gene expressions. The determination of opioid and cannabinoid receptor gene expression levels in pancreatic cancer cells and possibly also other cancer cells could thus provide important guidance on the selection of postoperative analgesia regimes and the prognosis of overall survival.

**Abstract:**

Pancreatic cancer (PDAC) has a poor prognosis despite surgical removal and adjuvant therapy. Additionally, the effects of postoperative analgesia with morphine and piritramide on survival among PDAC patients are unknown, as are their interactions with opioid/cannabinoid receptor gene expressions in PDAC tissue. Cancer-specific survival data for 71 PDAC patients who underwent radical surgery followed by postoperative analgesia with morphine (*n* = 48) or piritramide (*n* = 23) were therefore analyzed in conjunction with opioid/cannabinoid receptor gene expressions in the patients’ tumors. Receptor gene expressions were determined using the quantitative real-time polymerase chain reaction. Patients receiving morphine had significantly longer cancer-specific survival (CSS) than those receiving piritramide postoperative analgesia (median 22.4 vs. 15 months; *p* = 0.038). This finding was supported by multivariate modelling (*p* < 0.001). The morphine and piritramide groups had similar morphine equipotent doses, receptor expression, and baseline characteristics. The opioid/cannabinoid receptor gene expression was analyzed in a group of 130 pancreatic cancer patients. Of the studied receptors, high cannabinoid receptor 2 (CB2) and opioid growth factor receptor (OGFR) gene expressions have a positive influence on the length of overall survival (OS; *p* = 0.029, resp. *p* = 0.01). Conversely, high delta opioid receptor gene expression shortened OS (*p* = 0.043). Multivariate modelling indicated that high CB2 and OGFR expression improved OS (HR = 0.538, *p* = 0.011, resp. HR = 0.435, *p* = 0.001), while high OPRD receptor expression shortened OS (HR = 2.264, *p* = 0.002). Morphine analgesia, CB2, and OGFR cancer tissue gene expression thus improved CSS resp. OS after radical PDAC surgery, whereas delta opioid receptor expression shortened OS.

## 1. Introduction

Pancreatic ductal adenocarcinoma (PDAC) has a low 5-year survival of 5–7% [1]. Moreover, its median OS after surgery is just 18 months, and its incidence is rising [2,3]. Survival remains poor despite the use of multidisciplinary treatment strategies that combine surgical removal, adjuvant and neo-adjuvant chemotherapy, and radiotherapy [4,5,6].

For over a decade, many mostly retrospective studies have yielded conflicting results concerning the influence of postoperative analgesia on recurrence and survival for various cancers [7,8]. Mechanisms supporting the use of regional analgesia rather than potent opioids have been proposed [9], but several clinical studies have found no survival benefits for such approaches.

At the cellular level, our group and others have shown that the persistence of circulating tumor cells (CTCs) after radical cancer surgery is a negative prognostic factor [10,11,12]. We also showed that piritramide opioid analgesia reduces the presence of CTCs in colon cancer patients after surgery, potentially affecting survival [13,14], and that the expression of the cannabinoid-2 receptor (CB2) in cancer tissues improves survival in small-cell lung cancer [15]. Similar effects were also observed in hepatocellular cancer [16].

We therefore believe that the choice of postoperative analgesia may significantly affect survival in cancer patients. However, we also believe that these effects should be evaluated in relation to opioid and cannabinoid receptor expression in specific cancer tissues because the effects of individual opioids in cancer patients are likely to depend on the tumor’s unique receptor profile and biology.

Few studies on the influence of cannabinoid and opioid receptors on survival in PDAC have been reported. Cannabinoids mitigate cancer progression mainly by promoting apoptosis and autophagia via accumulation of the sphingolipid ceramide, which targets the stress-regulated protein p8 [17]. They also inhibit angiogenesis and invasiveness and have immunomodulatory properties [18,19,20,21]. Michalski et al. found that cannabinoid-1 receptor (CB1) and CB2 were expressed more strongly in PDAC cells than in normal pancreatic cells, and a survival analysis indicated that a low expression of CB1 in cancer cells was associated with better outcomes, while low levels of cannabinoid-metabolizing enzymes in cancer cells shortened survival [22].

With regards to opioid receptors, Zhang et al. found that high μ opioid receptor (OPRM) expression combined with a high perioperative dose of sufentanil was associated with significantly shorter OS and disease-free survival (DFS) in PDAC stages I-III, but high OPRM expression alone did not affect OS/DFS [23]. Zagon et al. showed that a native opioid peptide, opioid growth factor, suppressed the replication of PDAC cells in vitro in a dose-dependent manner via its receptor, OGFR [24]. Additionally, Haque et al. found that OPRM overexpression in murine and human PDAC cell lines increased proliferation and cancer stemness in vitro, whereas knocking down OPRM had the opposite effects. Moreover, direct morphine stimulation of OPRM and macrophages caused dose-dependent increases in proliferation, invasion, and levels of stemness markers in cancer cells, and morphine induced chemoresistance to chemotherapeutics used in PDAC treatment [25]. However, clinical studies on the role of perioperative morphine analgesia in PDAC progression following radical surgery are lacking.

The extent of radical PDAC surgery can make it challenging to provide adequate postoperative analgesia. In cases where perioperative epidural analgesia is not feasible, high doses of potent opioids are typically administered. This may be a problem if opioid receptor activity influences cancer development because there is evidence that patients are particularly vulnerable to cancer development during the perioperative period: Shakhar et al. showed that major surgery can suppress the anticancer immune response [26]. For decades, morphine has been the gold-standard agent for opioid analgesia worldwide. Piritramide, a potent opioid with a unique chemical structure, has only been used in a few European countries (i.e., Germany, Czech Republic, Netherlands, Austria) [27,28,29,30] and is thus is a less well-known option for postoperative analgesia. It is a 4-amino piperidine derivative (2,2,-diphenyl-4-[1-(4-carbamoyl-4-piperidino)-piperidine]-butyro-nitrile) [31] and has a relative potency of 0.75 compared to morphine [32]. It is typically administered parenterally to treat moderate to severe acute pain [33]. Given the effects of cannabinoid/opioid systems on cancer progression and the influence of morphine and piritramide on CTC levels following major surgery, we hypothesized that survival after radical PDAC surgery would differ between morphine and piritramide analgesia groups and may also depend on opioid and/or cannabinoid receptor expression in cancer tissues.

## 2. Materials and Methods

Data and samples representing 241 patients who underwent pancreatic cancer surgery at the University Hospital Olomouc and University Hospital Brno were biobanked between 2007 and 2020 and evaluated (Figure 1). Based on the inclusion and exclusion criteria (Table 1), the opioid receptor expression was analyzed in 137 tumor tissue samples. The comprehensive perioperative and follow-up data were mined in seventy-one patients, including the cause of death. Total doses of morphine and piritramide administered for postoperative analgesia were obtained, and piritramide doses were converted to morphine equivalents according to the equation 1 mg piritramide = 0.75 mg morphine [34]. Patients received either morphine or piritramide. No patients received both morphine and piritramide (Table 2). The quantitative real-time polymerase chain reaction was used to analyze the expression of the following opioid and cannabinoid receptors in tumor tissue samples: OGFR, OPRM, OPRD, kappa opioid receptor (OPRK), lambda opioid receptor (OPRL), CB1, and CB2.

### 2.1. Analysis of Opioid and/or Cannabinoid Receptor Expression in Tumor Tissues

Total RNA extraction from 20 to 40 mg tumor tissue samples fixed in RNAlater (ThermoScientific, Wilmington, DE, USA) was performed using the TRI Reagent (Molecular Research Center, Cincinnati, OH, USA) and chloroform (Sigma-Aldrich s.r.o, St. Louis, MO, USA). The resulting RNA was then resuspended in diethylpyrocarbonate (DEPC)-treated water (Ambion, Austin, TX, USA) according to the manufacturer’s instructions. The purity and concentration of the RNA were assessed using a Nanodrop ND 1000 instrument (ThermoScientific, Wilmington, DE, USA).

Reverse transcription was performed using 3 µg of total RNA with random primers (Promega, Madison, WI, USA), RNAsin ribonuclease inhibitor (Promega, Madison, WI, USA), and RevertAid H Minus M-MuLV Reverse Transcriptase (Fermentas, Vilnius, Lithuania) in a 30 µL reaction volume according to the manufacturer’s instructions. The cDNA products were then stored at −20 °C until qPCR analysis.

Quantitative RT-PCR reactions were performed in LightCycler 384 Multiwell plates (Roche, Basel, Switzerland). In each reaction, 50 ng of cDNA was mixed with LightCycler 480 DNA Probes Master (Roche, Basel, Switzerland) and the appropriate TaqMan Gene Expression Assay (Life Technologies, Thermo Fisher Scientific, Waltham, MA, USA; CB1: Hs01038532_m1; CB2: Hs00275635_m1; OPRK: Hs00175127_m1; OPRD: Hs00538331_m1; OPRM: Hs01053957_m1; OPRL: 00173471_m1; OGFR: Hs01071266_m1; ACTB: Hs99999903_m1) [15]. The volumes of the reagent mixture and the sample were 9 µL and 1 µL, respectively, and each sample was applied to the plate in four replicates. Plates were amplified by performing 50 cycles with a LightCycler 480 instrument using the temperatures and amplification times specified in the protocol supplied with the TaqMan Gene Expression Assays. ACTB (encoding actin β) was amplified as a reference gene. Fluorescence signals and cycle threshold values (CT) were evaluated using LightCycler 480 Software, ver. 1.1. ΔCT values were calculated by normalization against ACTB.

### 2.2. Statistical Analysis

Statistical analysis was performed using R, ver. 3.5.2 (Core Team, 2018). The significance threshold was *p* < 0.05. Specific cut-off values for opioid and cannabinoid receptor expression were determined using the maxstat() function (maxstat R package, v. 0.7–25), which estimates cut-points based on the maximally selected log-rank statistic (using overall and cancer-specific survival as an outcome variable). The expression of individual receptors was then classified as low (>cut-off) or high (≤cut-off) based on the cut-off values. Pearson’s chi-squared test, Fisher’s exact test, the Wilcoxon exact test, and the *t*-test were used to compare patient groups receiving different analgesic treatments and having different levels of receptor expression. Univariate survival analysis was performed using the log-rank test and Cox proportional hazard models. In multivariate Cox regression models, age and sex were used as adjusting variables and disease stage was used as a stratification variable.

## 3. Results

### 3.1. Opioid and Cannabinoid Receptor Gene Expressions’ Effects on Overall Survival

In total, 7 of the 137 patients initially included in the study had inconclusive receptor gene expression data, leaving 130 eligible for inclusion in the receptor gene expression analysis (Figure 1). Receptor gene expressions were categorized using estimated cut-off values (see Methods). All receptors were highly expressed in pancreatic tumor tissues except the OPRD (Figure 2).

The age, sex, and grading have no significant influence on opioid and cannabinoid receptor gene expressions in tumor tissues. The CB1 and CB2 receptor gene expressions in tumor tissues were significantly associated with disease stage (*p* = 0.013, resp. *p* = 0.002). The higher the stage, the lower the cannabinoid receptor gene expressions.

In patients with high CB2 receptor gene expression in tumor tissue, a significantly longer OS was found (log-rank test, *p* = 0.027; HR = 0.6, *p* = 0.028) (Figure 3). The multivariate Cox model analysis stratified by disease stage and adjusted for age and sex confirmed the findings (HR = 0.650; CI = (0.420–1.006); *p* = 0.053).

Patients with high OGFR receptor gene expression in tumor tissue had significantly longer OS (log-rank test, *p* = 0.009; HR = 0.631, *p* = 0.01) (Figure 4). This finding was confirmed by a multivariate Cox model analysis stratified by disease stage and adjusted for age and sex (HR = 0.588; CI = (0.372–0.927); *p* = 0.022).

On the contrary, the high OPRD receptor gene expression negatively affected overall survival (log-rank test, *p* = 0.041; HR = 1.6, *p* = 0.043) (Figure 5). The multivariate Cox model analysis stratified by disease stage and adjusted for age and sex confirmed the findings (HR = 1.655; CI = (1.012–2.707); *p* = 0.045).

An additional multivariate Cox model analysis with stepwise selection was performed that included the expression levels of all opioid and cannabinoid receptors stratified by disease stage and adjusted for age and sex. As shown in Table 3, this revealed that patients with high CB2 and OGFR receptor expressions had a significantly longer OS (HR = 0.538, *p* = 0.011, resp. HR = 0.435, *p* = 0.001), while those with high OPRD receptor expression had a significantly shorter OS (HR = 2.264, *p* = 0.002).

### 3.2. Postoperative Analgesia Effects on Cancer-Specific Survival

Of the 71 analyzed patients (31 female and 40 male, median age 63 years), 48 (67.6%) received morphine analgesia and 23 (32.4%) received piritramide analgesia in the postoperative period. The median morphine dose was 90 (70–120) mg and that for piritramide was 135 (82.5–180) mg, corresponding to a morphine equivalent dose of 101.2 (61.9–135) mg. The two groups thus had similar morphine equivalent dosage regimes and baseline characteristics (Table 2). The opioid and cannabinoid receptor expressions in tumor tissues were similar in both groups (Figure 6).

Patients receiving morphine analgesia had a significantly longer cancer-specific survival (CSS) than those receiving piritramide analgesia (22.4 vs. 15 months) according to a log-rank test (HR = 1.8, *p* = 0.04) (Figure 7). In a multivariate Cox model analysis stratified by disease stage and adjusted for age and sex, piritramide had a negative effect on CSS (HR = 2.904; CI = 1.485–5.679; *p* = 0.002) when compared to morphine.

An additional multivariate Cox model analysis with stepwise selection was performed that included the expression levels of all opioid and cannabinoid receptors as well as the applied analgesic treatment, stratified by disease stage and adjusted for age and sex. As shown in Table 4, this revealed that patients with high CB2 receptor expression had significantly longer CSS (HR = 0.186, *p* < 0.001), while those with high OPRD receptor expression had significantly shorter CSS (HR = 4.886, *p* < 0.001).

## 4. Discussion

We found that morphine analgesia improves cancer-specific survival (CSS) after radical PDAC surgery when compared to piritramide analgesia. Additionally, overall survival (OS) is increased by high CB2 and OGFR tumor tissue gene expression and reduced by high delta OPRD tumor tissue gene expression. Finally, all receptors were highly expressed in pancreatic tumor tissues except the OPRD. To our knowledge, this is the first study describing the survival benefits of morphine analgesia compared to piritramide analgesia after radical PDAC surgery and also the first study describing the effects of CB2 and OGFR gene expression on survival in PDAC patients.

Morphine analgesia has often been regarded as a negative factor that is associated with cancer recurrence. Several mechanisms describing its effects on cancer cells and anticancer immunity have been proposed to justify this position. Both in vivo and in vitro studies have shown that morphine enhances cancer cell proliferation, tumor progression, and cancer recurrence [35]. However, the evidence concerning its effects on cancer cell invasion [36,37] and angiogenesis promotion [38,39,40] is inconclusive and there are little data on its effects in PDAC. Zagon et al. found that OGF and OGFR suppressed the growth of PDAC in culture and in nude mice. Modulation of the OGF-OGFR pathway may potentially have a therapeutic effect in patients with pancreatic cancer [24,41,42,43]. Our results are consistent with those of Zagon et al. in that OGFR cancer tissue gene expressions improve overall survival in patients with pancreatic cancer. However, in our study, morphine analgesia proved beneficial for OS. There are several possible explanations for this discrepancy. First and foremost, our findings are based on data from a clinical setting involving major surgery. As such, they reflect the influence of several factors that are absent from controlled in vitro environments. In this context, it is notable that two recent prospective clinical trials examining lung and colon cancer surgery detected no survival benefits for regional analgesia when compared to morphine [8,44]. The effects of morphine analgesia in clinical cancer surgery thus remain unclear. Another important point is that our study compared morphine to piritramide and was not designed to determine whether morphine provides any survival benefit per se. However, since the median survival following PDAC surgery is only 18 months [2,3], but the OS of the morphine group in our study was 22.4 months, negative effects of morphine analgesia seem unlikely.

The effects of piritramide analgesia on cancer have received less attention than those of morphine. However, our group recently found that piritramide analgesia reduced CTC levels following colon cancer surgery [13,45] and could thus positively influence cancer recurrence/survival after major surgery. The fact that it had unfavorable effects on PDAC in the current study strongly suggests that individual cancer types differ in their molecular biology and responses to opioid and/or cannabinoid stimulation. Because our results indicated that high CB2 and OGFR tumor tissue gene expression improved OS, we hypothesize that piritramide’s negative effects may be linked to interactions with specific receptors in cancer and immune system cells. Unfortunately, studies on these interactions and comparisons between piritramide and morphine are lacking.

Our data on cannabinoid receptors revealed high CB2 gene expression in PDAC tissue, in accordance with the results of Michalsky et al. [22]. However, contrary to their findings, we observed that CB1 gene expression remained low, and OS was increased by high CB2 expression but unaffected by CB1 expression. These differences may be explained by the following observation made by Michalsky et al.: “cancer cells within single tissue samples showed various extents of CB1 and CB2 staining, ranging from no immunoreactivity to strong immunoreactivity”. Such broad variation within single tissue samples suggests that the results obtained depended heavily on the part of the sample that was chosen for analysis, rendering comparisons difficult. Therefore, studies using more reliable and reproducible tissue analyses are needed. Given that only very small portions of surgically removed tumors are usually analyzed, it is not clear that traditional methods can provide conclusive data on receptor expression. New types of analyses that can provide information on the majority of a tumor’s bulk may thus be needed. In addition, the number of samples tested in both studies was relatively low, which may partly explain the observed variation.

Because CB2 expression has been linked to longer survival in both lung [15] and hepatocellular cancer [16], similar mechanisms may be active in PDAC. In particular, CB2 is a likely target for endo- and exocannabinoids with anticancer activity resulting from effects on motility and migration, reductions in invasiveness and angiogenesis, and the induction of apoptosis [17,46]. Additionally, Michalsky et al. [22] observed that PDAC cells exhibit elevated cannabinoid-hydrolyzing enzyme activity, which could be interpreted as evidence of an adaptive mechanism to evade the anticancer effects of endocannabinoids. There is also emerging evidence of a complex interplay between cannabinoid and opioid receptors. Both receptor types are G-protein-coupled and they can form receptor heteromers containing both cannabinoid and opioid receptor proteins (exemplified by the CB1-OPRL heteromer). This considerably modifies their individual properties, and stimulation of one part of the heteromer may change the responsiveness of the other. This could have major clinical implications; combined therapies using low doses of drugs targeting cannabinoid and opioid receptors (or novel bivalent drugs) have shown promising results in humans [47]. Our observations of shorter OS in patients with high OPRD gene expression and longer OS in those with high CB2 and OGFR gene expression could thus plausibly be interconnected.

We recognize that our study has limitations. First, the number of patients included in this retrospective study was relatively small. Second, although the morphine and piritramide groups had similar baseline characteristics, data on the quality of postoperative analgesia based on validated scores were unavailable. Since there is some evidence that suboptimal pain management may promote cancer recurrence [48], a substantial difference in pain intensity between the groups, if present, could have influenced our findings. In addition, the total opioid doses in both groups were relatively small (Table 2) given the extent of open pancreatic surgery, which could raise questions about the adequacy of the analgesia. It should be noted that various non-opioid co-analgesics are generally used when treating PDAC patients. In this regard, it is important that several clinical trials have confirmed that NSAIDs have an opioid-sparing effect in patients who received opioids in the postoperative period [49,50]. The use of NSAIDs could influence results by reducing the total dose of postoperative opioids and thus reducing the adverse effects of opioids. In our study, the total equivalent doses of opioids were similar in both groups, so significant differences in the quality of analgesia or significant differences in the postoperative administration of NSAIDs are unlikely. However, since information on the administration of co-analgesics was not available for most of the patients, we cannot exclude differences in corticosteroid administration, which may be associated with improved survival in PDAC [3]. Third, it was not possible to reliably retrieve data on blood product administration, a factor that is likely to influence the recurrence of cancers including PDAC [3]. Nevertheless, based on the reviewed records, it can reasonably be assumed that a majority of patients in both groups received blood products during their stay in hospital.

We used cancer-specific survival for 36 months in our survival analysis because PDAC is characterized by rapid progression, which makes it difficult to accurately determine the timing of recurrence—the achieved accuracy depends on the frequency/quality of follow-up examinations, and recurrence may not be detected for some time after it emerges. In a systematic review, Petrelli et al. observed that OS and DFS were only weakly correlated in PDAC and concluded that OS should be the preferred survival endpoint because most patients die directly from the cancer and related complications [51].

To conclude, the 65% improvement in CSS observed for the morphine analgesia group (from 15 to 22.4 months) when compared to piritramide is important and should influence postoperative PDAC management if confirmed in other studies. Moreover, our findings regarding the influence of CB2, OGFR, and OPRD gene expression on OS may improve prognostication in PDAC and enable better personalization of care in the future. However, given the limitations of our study, we suggest that these results should be seen primarily as hypothesis-generating. Prospective studies are needed to elucidate the relationship between different types of postoperative analgesia and cancer recurrence/survival in PDAC. Molecular biological studies should also be conducted to clarify the role of opioid and/or cannabinoid receptors and their agonists/antagonists in cancer promotion. Ideally, the effects of individual opioids should be prospectively studied and related to the influence of the expression and/or activity of opioid and/or cannabinoid receptors in cancer and immune system cells. Finally, the role of opioid and cannabinoid receptor heteromers warrants investigation.

## 5. Conclusions

Morphine analgesia improves CSS compared to piritramide analgesia after radical pancreatic cancer surgery. Cannabinoid receptor 2 and opioid growth factor receptor are highly expressed in pancreatic cancer tissue and their high expression improves OS, whereas high delta opioid receptor expression reduces OS. More studies are needed to elucidate the effects of opioid treatment and the expression of opioid and cannabinoid receptors on the treatment of pancreatic cancer and to determine their prognostic value.

## Figures and Tables

**Figure 1 cancers-15-04038-f001:**
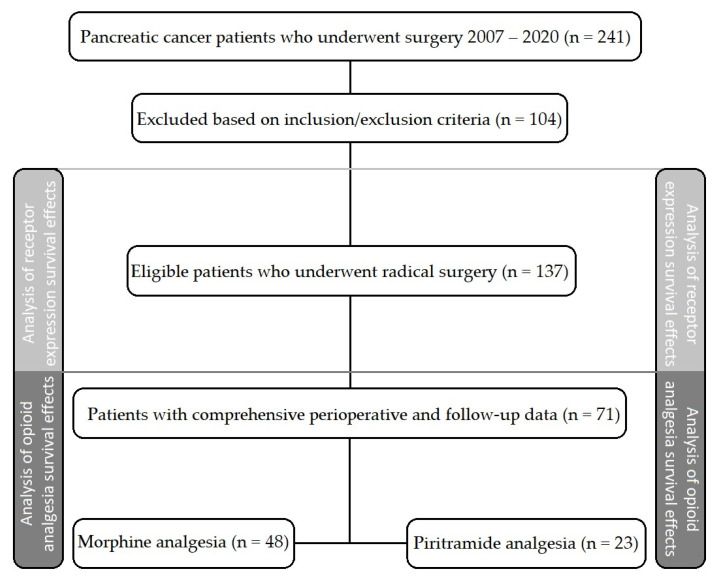
Study flow diagram.

**Figure 2 cancers-15-04038-f002:**
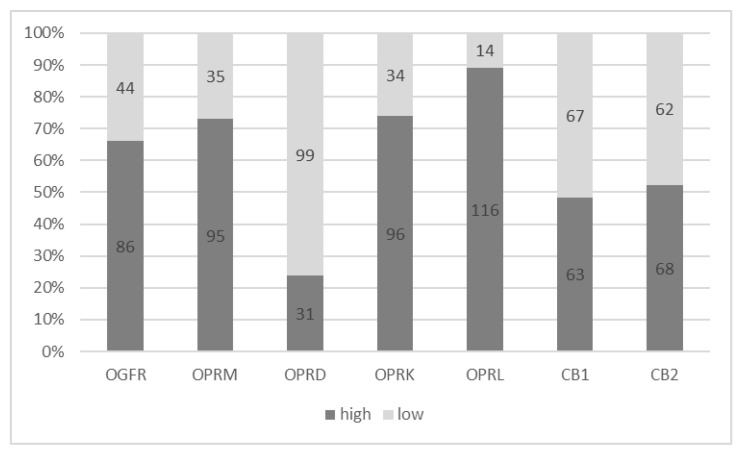
Opioid and cannabinoid receptor gene expressions. Abbreviations: OGFR = opioid growth factor receptor, OPRM = mu opioid receptor, OPRD = delta opioid receptor, OPRK = kappa opioid receptor, OPRL = lambda opioid receptor, CB1 = cannabinoid receptor 1, CB2 = cannabinoid receptor 2.

**Figure 3 cancers-15-04038-f003:**
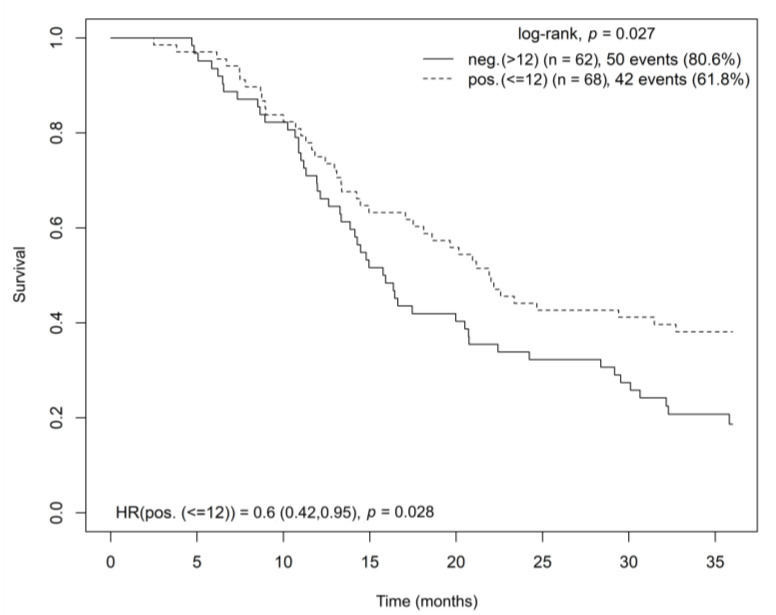
Kaplan–Meier curve showing overall survival for radically resected patients with pancreatic adenocarcinoma based on CB2 gene expression in tumor tissue. Abbreviations: HR = hazard ratio, CB2 = cannabinoid receptor 2.

**Figure 4 cancers-15-04038-f004:**
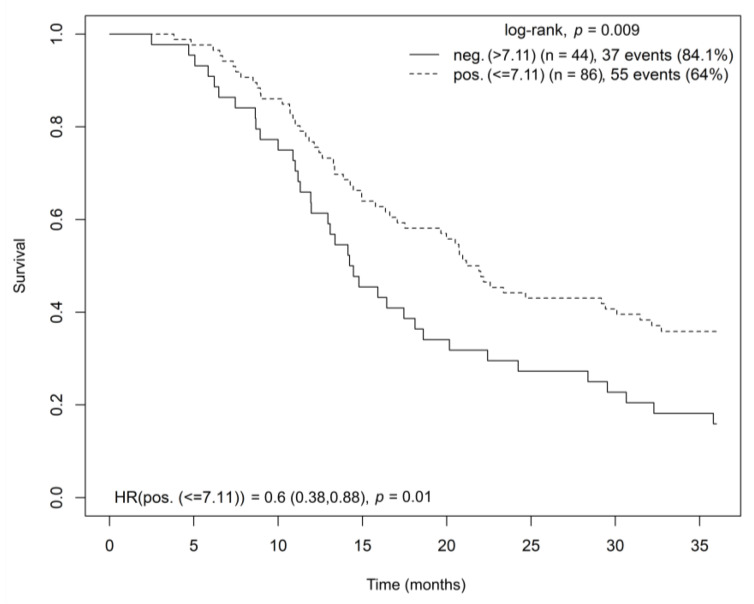
Kaplan–Meier curve showing overall survival for radically resected patients with pancreatic adenocarcinoma based on OGFR gene expression in tumor tissue. Abbreviations: HR = hazard ratio, OGFR = opioid growth factor receptor.

**Figure 5 cancers-15-04038-f005:**
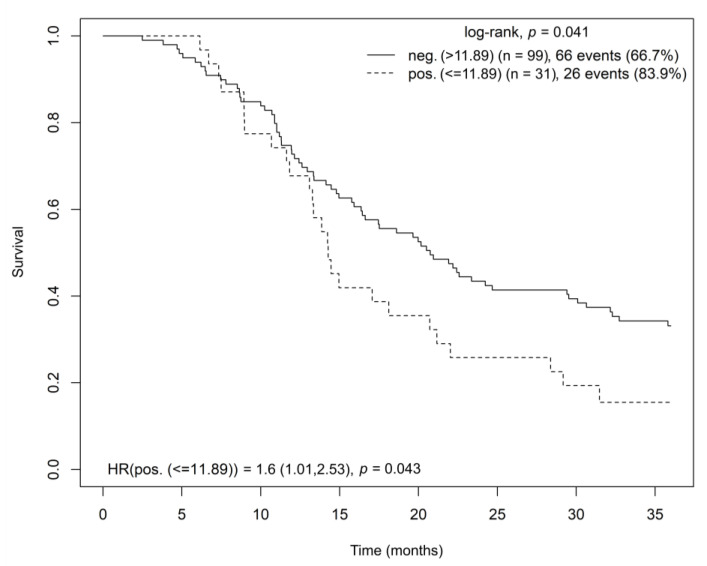
Kaplan–Meier curve showing overall survival for radically resected patients with pancreatic adenocarcinoma based on OPRD gene expression in tumor tissue. Abbreviations: HR = hazard ratio, OPRD = opioid receptor delta.

**Figure 6 cancers-15-04038-f006:**
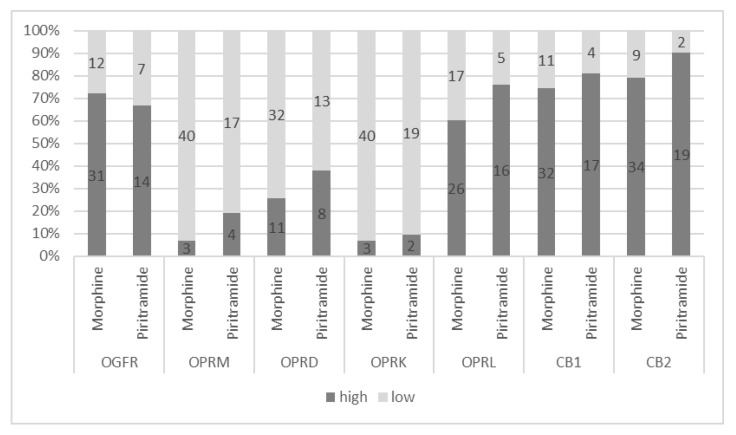
Opioid and cannabinoid receptor expressions in different opioid treatment groups. Abbreviations: OGFR = opioid growth factor receptor, OPRM = mu opioid receptor, OPRD = delta opioid receptor, OPRK = kappa opioid receptor, OPRL = lambda opioid receptor, CB1 = cannabinoid receptor 1, CB2 = cannabinoid receptor 2.

**Figure 7 cancers-15-04038-f007:**
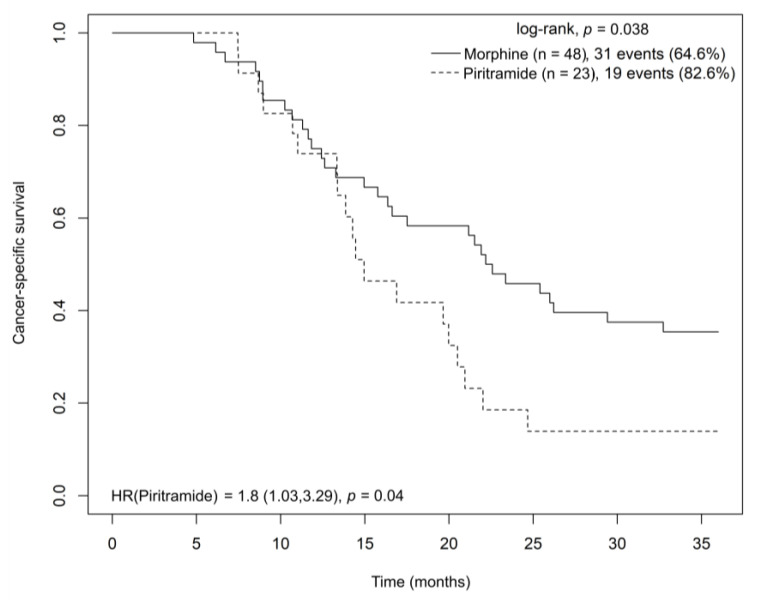
Kaplan–Meier curve showing cancer-specific survival for radically resected pancreatic adenocarcinoma patients treated with morphine or piritramide during the perioperative period. Abbreviations: HR = hazard ratio.

**Table 1 cancers-15-04038-t001:** Inclusion and exclusion criteria.

Inclusion Criteria	Exclusion Criteria
Age > 18 years	Tumor duplicity
Pancreatic adenocarcinoma (except ampullary carcinoma)	Death or reoperation within 30 days of surgery
Stage I, II, III (TNM classification)	
R0/R1 resection radicality	
Morphine or piritramide postoperative analgesia	

**Table 2 cancers-15-04038-t002:** Summary of the patients’ basic characteristics. Abbreviations: OS = overall survival, CSS = cancer-specific survival, NA = not available, CI = confidence interval.

Patient Characteristics	All Patients (*n* = 137) *n* (%)	Morphine (*n* = 48) *n* (%)	Piritramide (*n* = 23) *n* (%)
Sex			
Male	66 (48.5)	24 (50)	16 (69.6)
Female	70 (51.5)	24 (50)	7 (30.4)
Tumor stage			
I	14 (10.2)	7 (14.6)	7 (30.4)
II	109 (79.6)	40 (83.3)	16 (69.6)
III	14 (10.2)	1 (2.1)	0 (0)
Tumor grade			
1	6 (4.4)	2 (4.2)	4 (17.4)
2	78 (56.9)	26 (54.2)	11 (47.8)
3	53 (38.7)	20 (41.7)	8 (34.8)
Age (years)			
Median (q1–q3)	63 (58.5–69)	63 (58.5–69)	65 (58.5–69)
Resection			
R0	80 (58.4)	48 (100)	20 (87)
R1	57 (41.6)	0 (0)	3 (13)
Dosage (milligrams)			
Median (q1–q3)	NA	90 (70–120)	101.2 (61.88–135)
OS (months) Median (95% CI)	20.2 (16.4; 23.4)	22.4 (16.6; NA)	15.0 (13.4; 20.9)
CSS (months) Median (95% CI)	NA	22.4 (16.6; NA)	15.0 (13.4; 20.9)

**Table 3 cancers-15-04038-t003:** Multivariate Cox model analyzing the effects of all studied opioid and cannabinoid receptors on overall survival among patients with pancreatic adenocarcinoma. Abbreviations: HR = hazard ratio, CI = confidence interval, OGFR = opioid growth factor receptor, OPRM = mu opioid receptor, OPRD = delta opioid receptor, OPRK = kappa opioid receptor, OPRL = lambda opioid receptor, CB2 = cannabinoid receptor 2.

	HR	95% CI	*p*-Value
Age	1.011	0.987–1.036	0.363
Sex	1.057	0.677–1.65	0.806
OGFR	0.435	0.264–0.717	0.001
OPRM	2.076	1.199–3.594	0.009
OPRD	2.264	1.334–3.843	0.002
OPRK	0.480	0.286–0.805	0.005
OPRL	3.017	1.344–6.775	0.007
CB2	0.538	0.333–0.869	0.011

**Table 4 cancers-15-04038-t004:** Multivariate Cox model analyzing the effects of all studied opioid and cannabinoid receptors on cancer-specific survival among patients with pancreatic adenocarcinoma. Abbreviations: HR = hazard ratio, CI = confidence interval, OPRM = mu opioid receptor, OPRD = delta opioid receptor, CB2 = cannabinoid receptor 2.

	HR	95% CI	*p*-Value
Age	1.023	0.978–1.07	0.318
Sex	1.234	0.632–2.409	0.538
Piritramide	3.060	1.478–6.337	0.003
OPRM	0.203	0.055–0.750	0.017
OPRD	4.886	2.228–10.717	<0.001
CB2	0.185	0.079–0.435	<0.001

## Data Availability

The data presented in this study are openly available in FigShare at https://figshare.com/account/articles/21841071 (accessed on 6 August 2023) (DOI 10.6084/m9.figshare.21841071).

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
