# Peer review of "Morphine Analgesia, Cannabinoid Receptor 2, and Opioid Growth Factor Receptor Cancer Tissue Expression Improve Survival after Pancreatic Cancer Surgery"

_cancers, 2023, doi:10.3390/cancers15164038_

Round 1

Reviewer 1 Report

I was glad to review the work of the authors regarding this very interesting article on Morphine analgesia, cannabinoid receptor 2 and opioid growth factor receptor cancer tissue expression improve survival after pancreatic cancer surgery. Despite the major advances in postoperative analgesia, there are still numerous unanswered questions regarding the ideal combination of analgesics that should be given postoperatively. The manuscript is well-written and the incorporated tables and figures make the study easy to follow.

I strongly recommend acceptance for publication of the paper after minor changes.

1) According to the results of this study, morphine analgesia improves cancer-specific survival (CSS) after radical PDAC surgery when compared to piritramide analgesia. However, I would suggest a brief discussion on the notion of a significant opioid-sparing effect of NSAIDs in postoperative pain management after surgical operations.

Consider citing this randomized control trial:

https://pubmed.ncbi.nlm.nih.gov/33155461/ 

Author Response

Dear Reviewer 1, many thanks for your valuable suggestions.

  1. According to the results of this study, morphine analgesia improves cancer-specific survival (CSS) after radical PDAC surgery when compared to piritramide analgesia. However, I would suggest a brief discussion on the notion of a significant opioid-sparing effect of NSAIDs in postoperative pain management after surgical operations. Consider citing this randomized control trial: https://pubmed.ncbi.nlm.nih.gov/33155461/

Reply: Agreed. The short paragraph discussing the NSAID effect has been added including two new references. See page 13, lines 367 – 373.

Reviewer 2 Report

The authors analyze a topic which is of continuous interest – improve survival after pancreatic cancer surgery.

The presentation is clear, comprehensive and well documented.

How was it decided in which group a patient should belong to?   If a patient received morphine didn t he receive piritramide at all? and vice-versa…

The 4 tables offer information on the patients and statistical data. There are no data on the number of R0 and R1 resections in the 2 groups which may have influenced survival.

In row 98 instead of mu it should be μ which is the correct Greek letter pronounced approximately mu.

The references are appropriate, up-to-date and contain 48 titles.

The 7 figures are appropriate and mandatory for sustaining the topic.

I found no plagiarism.

The discussions and conclusions are coherent and connected to the content.

It is a retrospective study and contains a small number of patients per year (17) but brings new ideas.

In my opinion the paper fits the journal and the language is correct and understandable.

I recommend the paper to be accepted with the demanded corrections.

Author Response

Dear Reviewer 2, many thanks for your valuable suggestions.

  1. How was it decided in which group a patient should belong to? If a patient received morphine didn t he receive piritramide at all? and vice-versa…

Reply: The study was observational and retrospective. The patients received the analgesia randomly and were enrolled consecutively. Patients received either morphine or piritramide. No patients in our cohort received both morphine and piritramide.  See page 4, lines 139-140

  1. The 4 tables offer information on the patients and statistical data. There are no data on the number of R0 and R1 resections in the 2 groups which may have influenced survival.

Reply: Agreed. The type of resection was added into the Table  2. The effect of the opioid analgesia on cancer specific survival was analyzed mostly in R0 resected PDAC patients (68/71). See page 5, Table 2.

  1. In row 98 instead of mu it should be μ which is the correct Greek letter pronounced approximately mu.

Reply: Agreed and edited. See page 3, line 98.